# High spatial dynamics-photoluminescence imaging reveals the metallurgy of the earliest lost-wax cast object

M. Thoury[1,2], B. Mille[3,4], T. Séverin-Fabiani[1,2], L. Robbiola[5], M. Réfrégiers[2], J.-F. Jarrige[6,7,‡] & L. Bertrand[1,2]

Photoluminescence spectroscopy is a key method to monitor defects in semiconductors from nanophotonics to solar cell systems. Paradoxically, its great sensitivity to small variations of local environment becomes a handicap for heterogeneous systems, such as are encountered in environmental, medical, ancient materials sciences and engineering. Here we demonstrate that a novel full-field photoluminescence imaging approach allows accessing the spatial distribution of crystal defect fluctuations at the crystallite level across centimetre-wide fields of view. This capacity is illustrated in archaeology and material sciences. The coexistence of two hitherto indistinguishable non-stoichiometric cuprous oxide phases is revealed in a 6,000-year-old amulet from Mehrgarh (Baluchistan, Pakistan), identified as the oldest known artefact made by lost-wax casting and providing a better understanding of this fundamental invention. Low-concentration crystal defect fluctuations are readily mapped within ZnO nanowires. High spatial dynamics-photoluminescence imaging holds great promise for the characterization of bulk heterogeneous systems across multiple disciplines.

[1] IPANEMA, CNRS, ministère de la Culture et de la Communication, Université de Versailles Saint-Quentin-en-Yvelines, USR 3461, Université Paris-Saclay, 91128 Gif-sur-Yvette, France. [2] Synchrotron SOLEIL, 91128 Gif-sur-Yvette, France. [3] C2RMF, Palais du Louvre, 75001 Paris, France. [4] PréTech, CNRS, Université Paris Nanterre, UMR 7055, 92023 Nanterre, France. [5] TRACES, CNRS, ministère de la Culture et de la Communication, Université Toulouse—Jean Jaurès, UMR 5608, 31100 Toulouse, France. [6] ArScAn, CNRS, Université Paris Nanterre, Université Paris 1, ministère de la Culture et de la Communication, UMR 7041, 92023 Nanterre, France. [7] Institut de France, 23 quai de Conti, 75006 Paris, France. Correspondence and requests for materials should be addressed to M.T. (email: mathieu.thoury@synchrotron-soleil.fr).
‡ Deceased

For the last 15 years, specific cutting-edge developments have led to considerable improvements in photoluminescence-based analysis. Life sciences and semiconductor physics have been the main drivers strongly influencing instrumental choices[1,2]. In particular, monitoring target biomolecules with fluorescence imaging has led to major breakthrough in biomedical research[3]. A critical development has been specific antibody tagging, which provides the specificity and high quantum yield required to map and dynamically follow proteins within tissues at cellular level[4]. In solid-state physics, high-resolution low-temperature (helium) photoluminescence micro-spectroscopy has become the preferred technique to assess intrinsic electronic properties from individual nanostructures, such as the early state of chemical doping in single-walled carbon nanotubes[5]. Interpretation of spectral signatures collected at room temperature is challenging as emission bands are thermally broadened, particularly owing to the temperature-dependent phonon-coupling factors. Ultra high analytical sensitivity, great ease of use and emergence of super-resolved imaging have been instrumental to further establish photoluminescence as an essential tool in these fields. These optimizations have been driven by specific constraints; for instance, attaining nanoscale spatial resolutions has triggered near-field scanning at the expense of narrow fields of view and stringent requirements in sample surface roughness and slope. However, if major developments including near-field configuration, specific labelling and cryogenic environment have strongly enhanced the capability of characterizing specific biomolecules and semiconductor nanostructures, they are not directly applicable to imaging much of the very large range of mixed-compositional materials that are heterogeneous at bulk, such as those encountered in environmental, material, earth or planetary sciences, engineering and so on. In these samples, significant areas need to be studied at high spatial resolution to attain a statistically significant representation of materials' heterogeneity. Even for materials where specific staining would be applicable, it is often not an option owing to the alteration induced on the analyte. Characterization therefore needs to resort to autoluminescence. However, the high contrast in luminescence yields between intrinsic luminophores becomes a limiting factor. In addition, many samples cannot tolerate mechanical stress or chemical transformation induced by large temperature changes when placed in a cryogenic environment[6]. To tackle the characterization of such materials, the ideal system would allow covering all length scales from micrometric resolution to centimetres, providing wide tunability in excitation energy and detection from the deep ultraviolet to the near infrared to collect autoluminescent signatures, while being efficient at room temperature. Here we demonstrate the great benefit of gigapixel luminescence images obtained from coupling full-field imaging and optimized raster scanning. Versatile characterization of complex low-intensity photoluminescence signatures from crystallite sizes to whole macroscopic objects opens a new possibility for the study of polycrystalline semiconductors and other heterogeneous materials. For these materials, ensuring the best compromise between full tunability in excitation and emission, high spatial dynamics, that is, a high ratio between field of view and lateral resolution, and convenient room-temperature operation, is often more critical than reaching nanometric resolution. This means, for example, that we were able to study fluctuations in crystal defect density at the submicrometric scale while imaging this behaviour over centimetres. The wide tunability of the excitation, owing to the ability to switch between conventional and synchrotron sources, allows selecting an optimized excitation of luminophores above 200 nm.

We demonstrate this improved capability on two applications. Although use of advanced photoluminescence imaging has never been reported in archaeology, imaging reveals a hidden microstructure across a particularly challenging archaeological artefact. In a fully corroded 6,000-year-old small amulet identified as the earliest lost-wax cast and discovered in Mehrgarh (Baluchistan, Pakistan), one of the most important archaeological sites from the early Neolithic period, the clue to the entire metallurgical process of the earliest lost-wax cast amulet is provided by multiscale photoluminescence imaging. The methodology identifies the coexistence of two hitherto indistinguishable non-stoichiometric cuprous oxide phases and allows visualization of the spatial distribution of a ghost fossilized eutectic system, which reveals the innovative process they developed. All the images were collected on a fully customized synchrotron full-field microscope equipped with multispectral detection. The overall data cube results from the mosaicking of 414 tiles collected in three emission bands at three excitation energies, totalling 1.5 gigapixels. Using the same strategy, we could image structured crystal defects fluctuation within individual ZnO nanowires across populations of hundreds, from their low-yield photoluminescence. The continuous tunability of the synchrotron beam allows excitation down to the shortwave ultraviolet (UVC). We therefore demonstrate the exceptional potential of high spatial dynamics-photoluminescence imaging to study nano- and polycrystalline materials for applications within a variety of fields, ranging from quality control in semiconductor solid-state physics to geophysics, archaeology and environmental sciences.

## Results

**The Mehrgarh amulet is the earliest known lost-wax cast object.** To highlight the novelty of our approach, we report the information revealed by high spatial dynamics-photoluminescence imaging on a six-millennia old amulet discovered at Mehrgarh (Baluchistan, Pakistan), one of the most important archaeological sites from the early Neolithic period in the Ancient Near East (Fig. 1 and Supplementary Fig. 1).

The ornament with inventory number MR.85.03.00.01 was studied in detail (Fig. 1c,d). A visual inspection indicates that its 'spoked wheel' shape consists of six small rods lying on a ring of 20 mm diameter. At the centre of the wheel, the spokes were clearly pressed on each other until a junction was obtained by superposition; the base of each spoke was attached to the support ring using the same technique. Both the spokes and the support ring are circular in section. Only a wax-type material, that is, easily malleable and fusible, could have been used to build the corresponding models. This wheel-shaped amulet cannot result from casting in a permanent mould: this shape could not have been withdrawn without breaking the mould, as no plane intercepts jointly the equatorial symmetry planes of the support ring and of the spokes without inducing an undercut. The artefact was therefore cast using a lost-wax process (Supplementary Fig. 2).

A first campaign of measurements was performed 10 years ago but the wheel-shaped amulet could only be exhaustively described through novel advanced imaging. X-ray radiographs showed that it is corroded from its surface to its core. SEM examination of the equatorial section of the amulet corroborated the complete corrosion of the artefact, yet showed locally a fossilized dendritic structure, confirming a casting process. X-ray microanalyses on small areas highlighted Cu, O and Cl in the dendrites and Cu and O in the interdendritic space. Raman spectra allowed identifying the corrosion compounds: clinoatacamite $Cu_2(OH)_3Cl$ in the dendrite and cuprous oxide $Cu_2O$ in the interdendritic space. However, full corrosion of the metal to cuprous oxide $Cu_2O$ precluded any further understanding of the manufacturing and metallurgical processes.

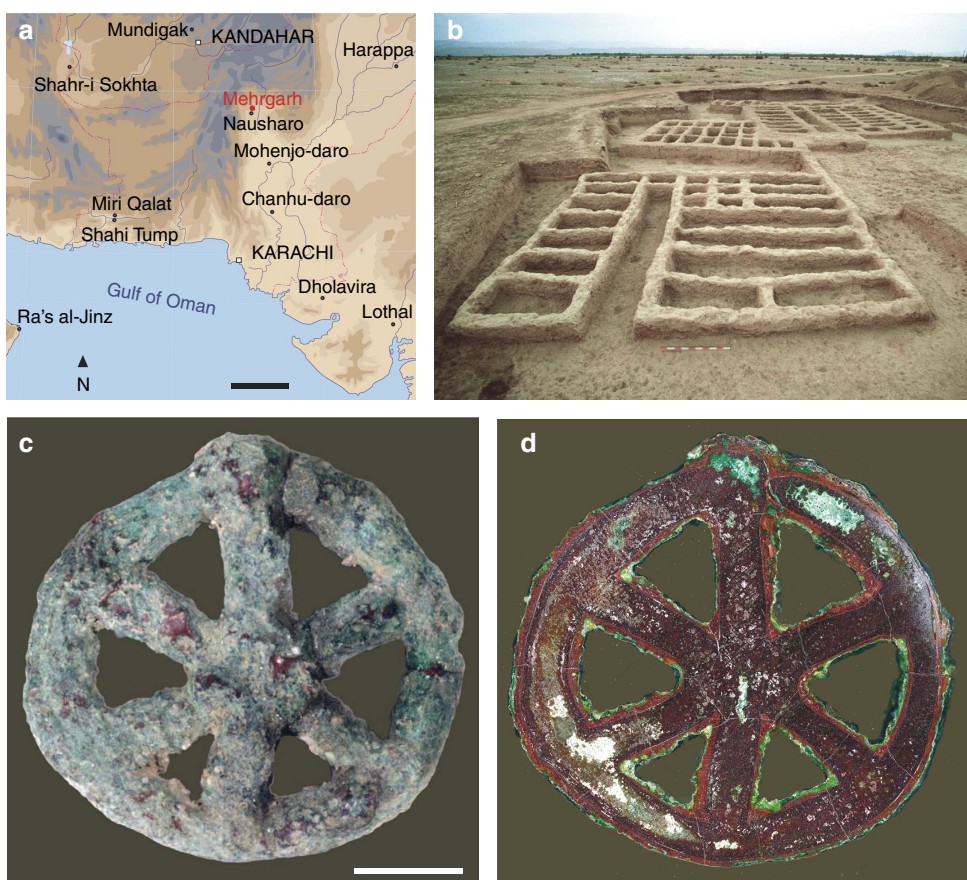

**Figure 1 | The amulet MR.85.03.00.01 from Mehrgarh.** (**a**) Map indicating the major Indo-Iranian archaeological sites dated from the seventh to the second millennia BC. Scale bar, 200 km. (**b**) View of the MR2 archaeological site at Mehrgarh (sector X, Early Chalcolithic, end of period III, 4,500–3,600 BC). (**c**) View of the front side of the wheel-shaped amulet. Scale bar, 5 mm. (**d**) Dark-field image of the equatorial section of the amulet.

**Macroscale imaging confirms casting in a single piece**. Photoluminescence imaging shows the continuity of the spatial distribution and orientation of the remnant dendritic structure all across the equatorial section (Figs 1d and 2a,b, Supplementary Fig. 3). This demonstrates that the artefact was cast in a single piece and does not consist of soldered parts (Supplementary Fig. 4). The lack of any crystal deformation shows that the object was made with very little, if any, subsequent work on the object, such as hammering. In addition, in the amulet three-dimensional morphology, no plane intercepts jointly the equatorial symmetry planes of the support ring and of the spokes without inducing an undercut. These observations therefore designate lost-wax casting as the procedure used for its fabrication. This is in agreement with the history of metallurgy in Baluchistan that shows evidence of an important development of lost-wax casting as demonstrated by finds such as the 'Leopards Weight', an extraordinary deco-rated ovoid ball of copper and lead weighing more than 15 kg dated end of the fourth millennium BC (ref. 7), and by the absence of any tradition of casting intricate shapes using piece-moulds as for instance reported in China[8].

**Mesoscale imaging reveals atypical metallographic structure**. Between corroded dendrites, hundreds of micrometres wide interdendritic spaces are observed in photoluminescence imaging. So-called 'ghost' dendritic structures are frequently observed in highly corroded ancient copper alloys[9]. On alloys, an interdendritic structure only occurs in the solidification of a two-phase system with alloying element such as Pb, As or Sn in ancient copper alloys.

Extensive investigation by optical microscopy, scanning electron microscopy with energy-dispersive spectroscopy (SEM-EDS) and Raman spectroscopy reveals no alloying element at the 100 μm length scale: red cuprous oxide $Cu_2O$ is ubiquitous in the extended interdendritic spaces, while green clinoatacamite $Cu_2(OH)_3Cl$ has formed in the corroded dendrites (Figs 2c,d and 3). The chemical composition of the interdendritic spaces is extremely homogeneous throughout the entire artefact (Fig. 3b–d, and Supplementary Fig. 4). Apart from copper and oxygen, only Ag and Fe are identified as traces with SEM-EDS ($< 0.2$ wt%, SEM-EDS). Synchrotron X-ray microfluorescence imaging over a spoke of the artefact detect, in addition, trace levels of Au, Ag and Hg in interdendritic spaces. The composition of the Mehrgarh artefact is therefore atypical, as copper was not alloyed with another metal. Electron backscatter diffraction (EBSD) performed at a submicron scale shows no other phase than cuprous oxide $Cu_2O$ within the interdendritic space (Supplementary Fig. 5).

**Microscale imaging reveals an invisible eutectic microstructure**. The intense photoluminescence signal within the interdendritic spaces appears to result from the presence of an exceptionally well-fossilized microscopic pattern, invisible with the other methods used (SEM, EBSD, white light OM, Raman spectroscopy). The $\sim 1$ μm lateral resolution allows the clear observation of a rod-like structure of high-yield luminescent $Cu_2O$ in the near infrared within a distinctly emitting $Cu_2O$ matrix (Fig. 2a,b). Such rod-like pattern, which has been preserved through corro-sion, is a direct signature of a eutectic growth. The interdendritic

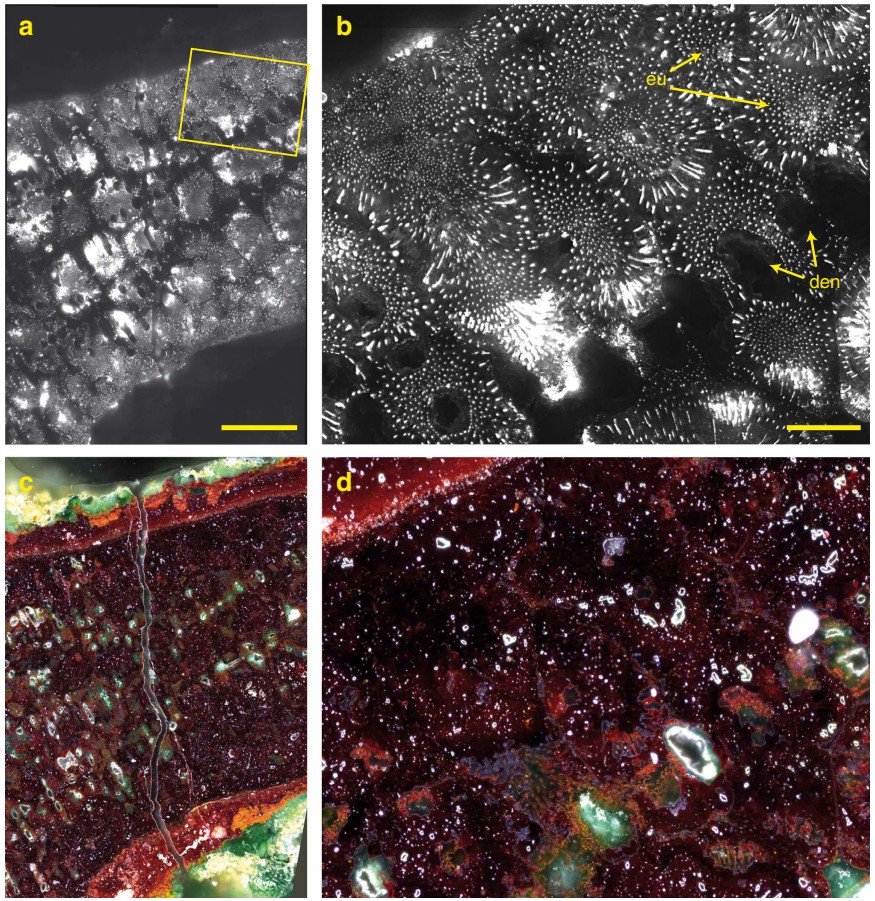

**Figure 2 | Fossil microstructure of the eutectic revealed in the 6,000-year-old Mehrgarh amulet.** Images reveal a typical eutectic morphology. The regular rod-like pattern is observed over millimetres in the interdendritic spaces. (**a**) Low magnification photoluminescence (PL) image of the wheel under 420–480 nm excitation and 850–1,020 nm bandpass emission ( ×40 objective, NA = 0.6). Scale bar, 500 µm. (**b**) Close-up view of the wheel ( ×100 objective, projected pixel size: 155 nm, NA = 1.25). den, dendrite; eu, rod-like eutectic in the interdendritic space. Scale bar, 100 µm. (**c**) Dark-field microscopy image of the same area of **a**. (**d**) Dark-field microscopy image of the same area of **b**. Note that the dendritic microstructure is more clearly evidenced in **a** than in **c**, and that the eutectic microstructure in **b** is not visible in **d**.

spaces therefore correspond to eutectic areas that were initially composed of $Cu^0$ with rod-like $Cu_2O$, and result from the hypoeutectic solidification of the binary system $Cu^0$–$Cu_2O$ in which initial $Cu^0$ dendrites were formed. During long-term corrosion at ambient temperature, the original $Cu^0$ has been oxidized to $Cu_2O$, while the rod-like eutectic $Cu_2O$ phase has been preserved. These two distinct cuprous oxides $Cu_2O$ observed today are hereafter designated as co-$Cu_2O$ (corrosion) and eu-$Cu_2O$ (eutectic), respectively. Strikingly, this micrometric structure was completely preserved over centimetres during six millennia (Supplementary Fig. 3). Due to the aggressive role of chlorides in the archaeological soil, dendritic $Cu^0$ was more affected by corrosion than eutectic $Cu^0$ in contact with eu-$Cu_2O$, inducing the progressive formation of $Cu_2(OH)_3Cl$ in the dendrites[11–13].

Pure $Cu_2O$ is a semiconductor whose spectroscopic properties are highly sensitive to intrinsic or extrinsic crystal defects[14,15]. Although uniquely consisting today of $Cu_2O$ (Fig. 3b–d), the different nature of atomic-scale crystal defects within eu-$Cu_2O$ and co-$Cu_2O$ of the interdendritic spaces allows visualization of the 6,000-year-old metallographic structure. The associated photoluminescence signal of the eu-$Cu_2O$ is dominated by emission in the near infrared from copper vacancies ($V_{Cu}$), while the excitonic emission near the band-edge transition at 2.1 eV is quenched[16,17]. The formation of eu-$Cu_2O$ at high temperature (the eutectic reaction occurs at 1,066 °C, Supplementary Note 1), must have led to the creation of a high density of stable $V_{Cu}$.

**The oldest lost-wax cast.** The ability to cover all length scales continuously from crystallite sizes to macroscopic sample dimensions allows deciphering invisible patterns that provided the key for a complete understanding of the manufacturing of the Mehrgarh artefact. From the visual inspection of the artefact, we show that the 20 mm wheel-shape model was prepared in a waxy material: the spokes were brought together by pressing each other at the wheel centre, and the base of each spoke was pressed on the support-ring (Fig. 4a, Supplementary Fig. 2). Once made, the wax model was invested into a clay mould. The clay mould was heated upside down to run out the wax; baking was extended at higher temperature to harden the mould and drive out any moisture. Copper was poured in the mould, taking the place of the wax to cast the artefact in a single piece (Fig. 4b). The absence of any alloying element or significant impurity except low traces of Au, Hg and Ag in the amulet points to the use of a very pure copper, possibly native copper, that was melted in air above 1,085 °C. Had arsenic been present, as in most coeval cast alloys known so far[18], the eutectic could not have formed, as oxidation of liquid copper is mitigated by the greater affinity of arsenic for oxygen[19]. The $Cu^0$–$Cu_2O$ phase diagram can be exploited to trace the metallurgical sequence. During casting, the furnace atmosphere was inevitably oxidizing, and the copper melt absorbed ∼0.3 wt% of oxygen (∼1.1 at.%, Supplementary Fig. 6 and Supplementary Note 1), leading to the observed hypo-eutectic structure. The solidification of the dendrites started at about 1,070–1,074 °C

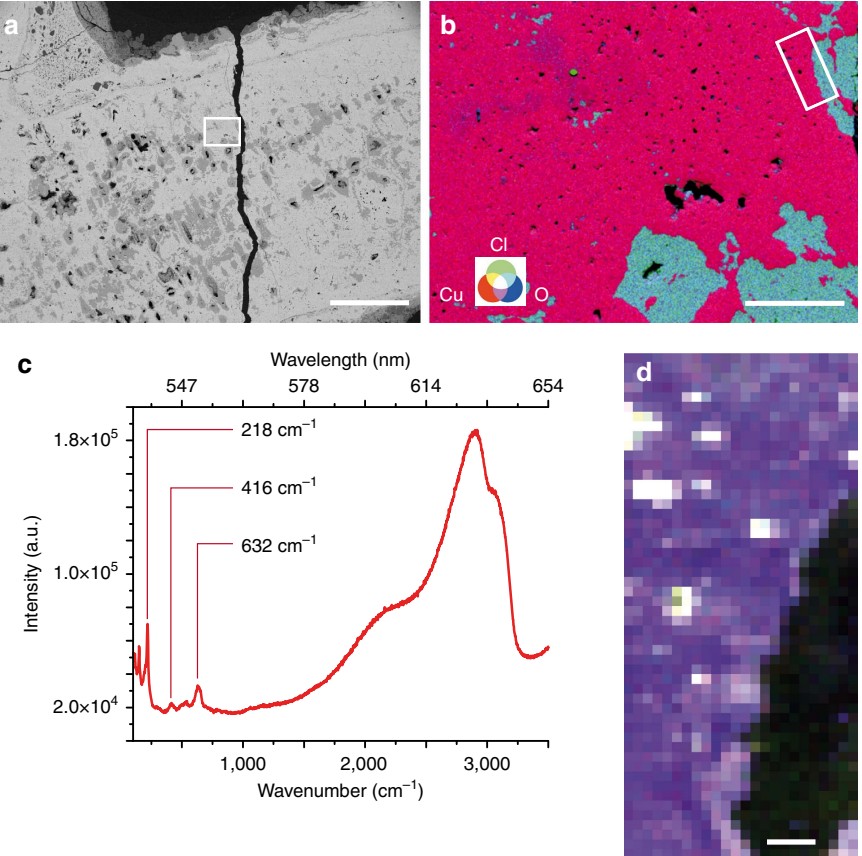

**Figure 3 | Mapping of Cu$_2$O species in interdendritic spaces.** (**a**) Image of dendrites and homogeneous interdendritic spaces (SEM-BEI, 10 kV). Scale bar, 300 μm. (**b**) RGB false colour image (SEM-EDS) of Cu (red), Cl (green) and O (blue) from the area denoted by a rectangle in **a**. Interdendritic spaces contain only Cu and O as major elements, while Cl is found in the corroded dendrites. Scale bar: 30 μm. (**c**,**d**) Identification of Cu$_2$O in interdendritic spaces in the area denoted by a rectangle in **b**. (**c**) Typical Raman spectrum from a Cu$_2$O region. The spectrum was obtained by averaging 12 scans within the zone imaged in **d** (using four pixels in three separate areas). (**d**) RGB false-colour image of Raman vibrational bands characteristic of Cu$_2$O: 632 (red), 416 (green) and 218 cm$^{-1}$ (blue). Raman spectroscopy mapping does not show any variation in the characteristic vibrational features of Cu$_2$O that would allow evidencing the rod-like eutectic structure. Scale bar, 4 μm.

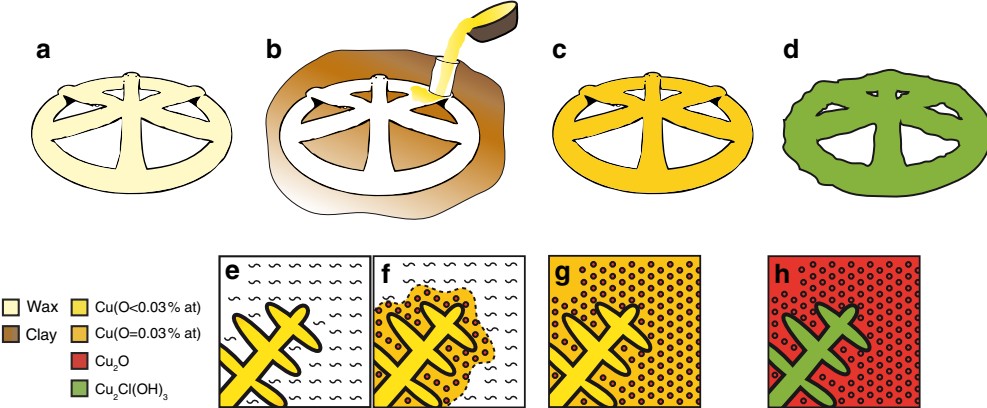

**Figure 4 | Manufacturing of the amulet from Mehrgarh by the lost-wax casting process.** (**a**) The model was shaped by manufacturing small rods circular in section in a very ductile material that melts at low temperature, such as beeswax. Each wax piece was welded to the other by a slight heating of their extremities. (**b**) The wax model was invested by a clay mixture to form a mould. The mould was heated to run out the wax, and copper was poured in the mould, taking place of the wax. (**c**) The final copper artefact was extracted by breaking the mould after cooling. (**d**) Totally corroded artefact after its 6,000-year burial. (**e**–**h**) Schematic representation of the solidification process and its evolution at a microscale: (**e**) 1,085 °C > T > 1,066 °C. Dendritic growth of metallic copper (oxygen content in dendritic Cu < 0.03% at). (**f**) Formation of the Cu-Cu$_2$O eutectic at 1,066 °C. The liquid phase solidifies into Cu$^0$ (0.03%at O) and a eu-Cu$_2$O rod-like structure. (**g**) Final metallurgical structure of crystals of dendritic copper (low in oxygen) surrounded by oxygen saturated Cu$^0$ (Cu$_{0.97}$O$_{0.03}$) and rod-like Cu$_2$O. (**h**) Current state of the artefact with the formation of the Cu$_2$Cl(OH)$_3$ phase within dendrites, while Cu$^0$ fully oxidizes to co-Cu$_2$O within the eutectic. eu-Cu$_2$O is fully preserved.

(Fig. 4e, Supplementary Fig. 6) while the eutectic formed at 1,066 °C (Fig. 4f). After cooling, the mould was broken and the casting was finished by cold working such as cutting the sprue and polishing (Fig. 4c,g). After burial, slow alteration took place in a sandy clayey soil and in a relatively dry environment (Fig. 4d,h; Supplementary Fig. 7). The ghost fossilization of the metallographic structure took several centuries to complete in a comparatively dry environment—at typically about one micrometre per year[20,21]—leading to a final uniform presence of $Cu_2O$ within the eutectic.

## Discussion

The discovery of the wheel-shaped amulets from Mehrgarh is an extraordinary evidence of the first attempts to manufacture precision casts by a lost-wax process. This innovation did not replace casting in permanent moulds but engendered a novel lineage of objects, whose complex shapes can only be obtained by this method. We can now state not only that metallurgists invented a totally new technique for casting, but also that control of the metal composition was part of their innovative research. By choosing a very pure copper rather than the usual arsenical copper[22], they used a metal whose origin was probably considered to be of higher value and quality. The traces of mercury, silver and gold identified in the corroded amulet form a typical pattern for native copper[23]. The use of high-purity copper turned out to be a dead end: this did not improve the casting properties of the melt but caused unfamiliar problems to the founder: the melting point is not decreased, whereas the metal castability is severely reduced[24]. Although the lost-wax process proved to be an irrefutable and permanent success, selecting very pure copper for casting has not been retained as a valid innovation. Looking for improvements, Baluchistan founders soon discovered that the addition of a large proportion of lead to copper (Pb: 10–30 wt%) vastly increased the metal fluidity. During the fourth millennium BC and up to the end of the third millennium BC, this new Cu–Pb alloy was extensively used, and solely dedicated for lost-wax casting[7,25]. Lost-wax casting and Cu–Pb alloy were therefore widely adopted in the Ancient Near East, and used to manufacture artefacts of the highest symbolic and ceremonial significance. The use of Cu–Pb alloy was only challenged at the beginning of the second millennium BC, when Cu–Sn bronze became widely used within this geographic area owing to its improved metallurgical properties.

Mehrgarh is a crucible for technological innovation during Neolithic and Chalcolithic times in the ancient South Asia from lithics, pottery, ornaments, clay figurines, glazed materials as well as textiles and early practice of dentistry[25–27]. The emergence of the lost-wax technique at Mehrgarh could have been triggered by several factors. The availability of beeswax is attested in the Near East at this period[28]. Second, recent works have proposed that lost-wax casting has been adopted more for the central role of beeswax as a ritually important material than for a technical need[29]. It is also significant that the very first objects made by lost-wax casting did not fully exploit the potential of lost-wax casting. The amulet here in question is practically flat, and arguably a rather similar one could have been cast more easily using an open mould. The wax rods used to shape the metal amulet closely resemble the small clay coils used to model hundreds of clay figurines and amulets discovered in the Neolithic and Chalcolithic levels of Mehrgarh, and possibly associated with a magical and/or religious function. With lost-wax casting, it was now possible to produce these traditional adornment artefacts in metal, by simply working wax in place of clay, maintaining the long-established way in which they were modelled. The specific context at the site

(resources, ritual, know-how) nurtured metallurgical invention, while other sites, possibly contemporaneous, such as Nahal Mishmar in the Levant that may have led to independent invention of lost-wax casting[30] did not provide the incubating context allowing dissemination to the entire ancient Near East. Lost-wax casting tested for the first time with the Mehrgarh artefact is still the premier technique for art foundry. It is also today the highest precision metal forming technique—under the name 'investment casting'—in aerospace, aeronautics and biomedicine, for high-performance alloys from steel to titanium[31]. Today, rapid prototyping technique such as three-dimensional printing offers revolutionary capabilities to design plastic, polymer or wax models used in investment casting[32,33]. New templating approaches for nanocasting semiconductor structures are among the latest evidence of the fundamental character of the lost-wax concept[34,35].

We demonstrate the potential of gigapixel photoluminescence imaging to study the response of materials at micrometric resolution over centimetre-size fields within desired spectral bands. The exploration of the spatial distribution of the electronic density of state within polycrystalline semiconductor materials is then possible. The proposed approach goes far beyond collection of point or average luminescence signal of great complexity, towards determination of the representative elementary areas in which the measured photoluminescence response in a heterogeneous matrix becomes continuous quantities. Here, high-definition images of crystal defect contrasts provide a direct probe of stoichiometry fluctuations, which in turn record information on the materials' manufacturing process. This approach can conversely prove to be extremely effective in optimizing the synthesis route of systems that are far less expected to be heterogeneous, such as batches of semiconductor nano-structures. We have therefore extended our proof of concept to a modern synthetic material by mapping and characterization of crystal defects density within a batch of nanowires. High signal-to-noise ratio images of zinc oxide nanowires of 0.5–1 μm in diameter and 14 μm in length deposited on a substrate were collected in nine spectral bands ranging from the deep ultraviolet to the near infrared using an excitation wavelength of 275 nm. The images reveal both unexpected spectral-dependent spatially variable emission from crystal defects along the length of individual nanowires and the statistical variability of the distribution of those defects within the entire population where a limited number of typical nanowire behaviours is observed (Fig. 5). Deep ultraviolet-optimized multispectral collection strategy allows 'à la carte' adaptation of integration times to each spectral emission range, to collect extremely low-yield responses that would otherwise go undetected through hyperspectral data collection. The ability to collect emission from single grains or crystallites to centimetres of samples at room temperature with tuneable source over the whole deep ultraviolet to near infrared range therefore provides unprecedented capability to image the intrinsic complexity of heterogeneous materials from nanosciences, engineering, geophysics, archaeology and environmental sciences.

## Methods

**Photoluminescence imaging.** Photoluminescence micro-imaging was performed on a full-field inverted microscope (Axio Observer Z1 microscope, Zeiss) at the DISCO beamline (SOLEIL synchrotron)[36]. The microscope is equipped with custom quartz lenses instead of the original glass ones, to ensure transmission of excitation and emission above 80% and allow collecting luminescence images down to 200 nm. The beamline exploits the tunability of the bending magnet source, with an energy bandwidth $\Delta E/E$ of $2 \times 10^{-2}$ at 275 nm (100 grooves per millimetre grating, iHR320 monochromator, Jobin-Yvon, Longjumeau, France).

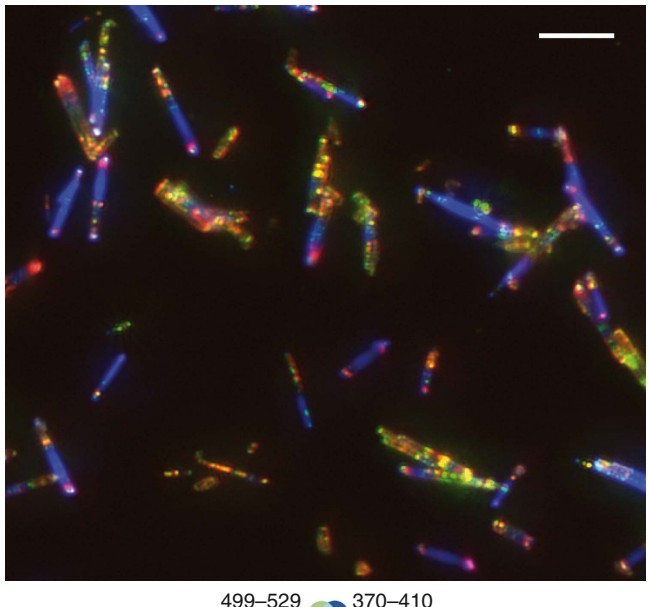

499–529 370–410

850–1,020

**Figure 5 | Spatial distribution of crystal defect and band edge emission of ZnO nanowires.** Full-field photoluminescence image of a batch of ZnO nanowires (ultraviolet excitation: 275 nm, 4.50 eV). False colour overlays of signal in the 850–1,020 nm (red), 499–529 nm (green) and 370–410 nm (blue) bands. The image is corrected in each channel from collection time, quantum efficiency of the CCD camera, transmission of emission filters and theoretical point spread function of the objective. Scale bar, 10 μm.

In the frame of this work, specific developments were implemented to optimize excitation tunability, high-throughput detection and spatial dynamics required to detect and spatially resolve the multi-scale luminescence pattern in the amulet (Supplementary Fig. 8a,b in comparison with Supplementary Fig. 8c–h). Two sources were coupled to attain the respective excitation ranges 220–400 nm (synchrotron radiation source)[36,37] and above 400 nm (halogen lamp coupled to an interference bandpass filter). In the deep ultraviolet (synchrotron) range, energies greater than 1.2 eV are blocked using a cold finger of thickness 7.5 mm that intercepts a vertical angle of 1.5 mrad in the middle of the beam. As a result, the spatial distribution of the beam at the exit of the monochromator is composed of two longitudinal sheets. To obtain a homogeneous field of illumination down to the deep ultraviolet, an optical set-up using micro-array lenses and a rotating diffuser was developed and positioned ahead of the microscope.

High-grade optical elements were used all along the optical path to minimize all optical distortions, particularly field and chromatic aberrations, and allow image stitching. A ×40 / NA 0.6 and ×100 / NA 1.25 glycerine Zeiss ultrafluar apochromatic immersion objectives were used to excite and collect images from ultraviolet-C to near infrared ranges. High spatial dynamic images were gathered by collecting mosaics of tiles with an XY motorized stage (PI) allowing to image areas of hundreds of micron side. For instance, Fig. 2a is made of overlapping tiles, each of $1.4 \times 10^4$ μm$^2$ (774 × 759 pixels), in a 414 images matrix that creates a final 4.0 mm$^2$ image (14,888 × 11,415 pixels). The projected pixel size of 155 nm is 2.4 times smaller than the theoretical diffraction limit of 374 nm ( = 935 nm/2/1.25) at 935 nm. Measurement of the optical point spread function across an ∼400 nm CdS particle shows that spatial resolution is ∼1 μm (Supplementary Fig. 9). During the optimization procedure of our set-up, the experiment was replicated four times on the amulet. For each measurement, the eutectic pattern could clearly be visualized in the images collected in the near infrared (Supplementary Fig. 10). In addition, all the tiles collected showed a similar reproducible pattern.

High-throughput spectral detection from UVC up to near infrared was achieved by using a multi-spectral detection using high-transmission interferential bandpass filters positioned in front of a back-illuminated 1,024 × 1,024 pixels CCD (PIXIS:1024BUV, Princeton Instrument 13 × 13 μm$^2$ pixel size)[38]. The images shown in this work were collected using 370–410, 499–529 and 850–1,020 nm interference bandpass filters (transmission >90%). The collection time is adjusted for each set of excitation/emission conditions to optimize the signal-to-noise ratio (up to a few minutes per tile).

**Optical microscopy.** Dark-field microscopy was performed using a Zeiss Axio Imager M2m microscope coupled to an AxioCam ICc5 camera, with ×5 and ×20

objectives (C2RMF). The images collected on an XY motorized stage were mosaicked to cover a large field of view.

**Raman spectroscopy.** Raman spectroscopy was performed at an excitation of 532 nm and on-sample power of 2 mW with a ×100 objective (SOLEIL, SMIS). The spectra were collected using an integration time of 2 s, accumulation of two spectra per point and a 25 μm spectrograph aperture slit.

**Scanning electron microscopy.** SEM and EDS were performed on a Zeiss Supra 55 VP coupled to a Bruker EDS system (Quantax 800, 30 mm$^2$ silicon drift detector (SDD); IPANEMA).

**Electron backscatter diffraction.** EBSD was conducted on a JSM 7100F apparatus equipped with an Oxford AztecHKL and NordlysNano with 4 FSD detector (Centre de Microcaractérisation Raimond Castaing, Toulouse, France). For this analysis, the surface was prepared using vibratory polishing (Buehler VibroMet 2, ChemoMET polishing cloth) with 50 nm colloid alumina suspension. A carbon coating a few nanometres was applied (Leica EMACE600). The experiments were performed at 20 kV (70° tilt) and data were processed using the Channel 5 Tango software.

**Sample preparation.** The wheel-shaped amulet inventory number MR.85.03.00.01 was collected in 1985 at the MR2 site of Mehrgarh during the excavations of the 'Mission Archéologique de l'Indus' (dir. Jean-François Jarrige) in collaboration with the Department of Archaeology and Museums of Pakistan. A section was prepared in the equatorial plane, embedded in epoxy resin (Epofix, Struers) and polished with diamond pastes up to 0.25 μm grain size (C2RMF).

**Preparation of the ZnO nanowires.** ZnO nanowires were grown at 850 °C by metal–organic chemical vapour deposition (MOCVD) on a (0001) sapphire substrate using diethylzinc and nitrous oxide as zinc and oxygen precursors (GEMaC, Versailles, France).

**Data availability.** The data that support the findings of this study are available from the corresponding author upon reasonable request.

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

## Acknowledgements

This article is dedicated to the memory of Jean-François Jarrige (1940–2014), former director of the musée Guimet in Paris, who discovered Mehrgarh in 1974 and directed the 'Mission Archéologique de l'Indus' from 1975 to 2014. Claudie Josse is warmly acknowledged for providing EBSD results (Centre de microcaractérisation Raimond Castaing, CNRS UMS 3623, Toulouse, France). We acknowledge SOLEIL for provision of synchrotron radiation under projects no 20120848 and 20130920. We thank Christophe Sandt at the SMIS beamline for access to Raman microscopy (SOLEIL synchrotron), Pierre Gueriau for complementary synchrotron XRF mapping (IPANEMA) and Frédéric Jamme (SOLEIL synchrotron) for providing support to generate the point spread function (PSF). We thank Pierre Galtier, Alain Lusson and Vincent Sallet (GEMaC UMR8635) for preparing and providing the ZnO nanowires. We thank Sebastian Schoeder (synchrotron SOLEIL) for the representation of the amulet in three dimensions. We especially thank Catherine Jarrige, Gonzague Quivron, Aurore Didier and Jérôme Haquet who provided complementary information about the metal artefacts from Mehrgarh. We thank Barbara Berrie, Catherine Perlès, Denis Gratias and Uwe Bergmann for critical re-reading of the manuscript.

## Author contributions

M.T. and L.B. designed the experiments. L.B., M.T. and T.S.-F. coordinated and drafted the manuscript. T.S.-F., M.T., L.B., B.M. and L.R. wrote the manuscript and prepared the figures. B.M. selected the artefact and provided the archaeometallurgical interpretation. L.R. provided the corrosion interpretation. The experiments and data analysis were performed at the DISCO beamline at synchrotron SOLEIL (M.T., T.S.-F., L.B., M.R.), SEM-EDS (L.R., B.M.), Raman (M.T., L.R.), EBSD (L.R.) and OM (B.M., T.S.-F.). J.-F.J. provided the archaeological information.

## Additional information

**Competing financial interests:** The authors declare no competing financial interests.

**Publisher's note**: 

