## [Peer Review File · Nature Communications]

Reviewers' comments:

Reviewer #1 (Remarks to the Author):

Ultimate spatial dynamics-photoluminescence imaging of crystal defects fluctuation exemplified on a 6000-year-old archaeological artifact

Peer-review by Marcos Martín-Torres

This manuscript presents a well-written report on a rigorous and original analytical study of what arguably represents the earliest metal object cast using the lost-wax technique. The paper has two main aims:

a) to present ultimate spatial dynamics-photoluminescence imaging (USD-PL) as a novel technique with great potential for the characterisation of bulk heterogeneous materials at multiple scales;

b) to demonstrate that a small, corroded object recovered in an early context at the site of Mehrgarh was manufactured by the lost-wax casting method, and therefore represents the earliest evidence for the use of this technique.

Both of the above points are covered convincingly and are significant for a wide range of disciplines, from materials science through to archaeology and archaeometallurgy. Therefore I believe the paper merits publication in Nature Communications, although I recommend a few revisions. These are noted below, together with some more general comments.

a) USD-PL

Although I have experience in the analysis of archaeological metal objects (including some highly corroded like the one discussed in this paper), and have used the other techniques employed in this project (optical microscopy, Raman and SEM), I do not have direct experience with photoluminescence spectroscopy. Hence other reviewers will be more qualified to comment on the merits of this technique, and the suitability of the protocols followed.

The main breakthrough demonstrated in this study is the step forward from traditional PL analysis enabled by the generation of gigapixel luminescence images; this is obtained by combining full-field imaging with optimised raster scanning, and in turn it allows analysts to map samples at successive length scales across several orders of magnitude, from micrometers to centimetres. The authors describe this as a 'paradigm shift'. While only time and future tests will tell whether USD-PL truly brings a new paradigm, it is safe to say that this development does increase very noticeably the versatility and potential applications of the technique.

The main demonstration of the proof of concept is the study of the corroded copper object from Mehrgarh. Figure 2 beautifully illustrates how USD-PL allows the analyst to see ghost microstructural features that would be invisible under conventional light microscopy. The ghost eutectic structure of the non-stoichiometric copper oxides is revealed distinctively, and proves beyond reasonable doubt that the object was cast in one piece.

While the results are thus quite convincing, it would be interesting to compare these two sets of images (optical microscopy and USD-PL) to images of the same area obtained under an SEM. The latter is a technique commonly employed by archaeometallurgists to examine ghost microstructures in corroded objects, and one is left wondering if the features visible by PL would not have been observed under an SEM too. Admittedly, an SEM image is unlikely to be as impressive as that of PL, not least because of the contrast in luminescence that would not be perceptible by a BSE or SE detector on an SEM. However, a comparison between SEM and USD-PL would perhaps allow a more realistic graphic assessment of the differences between both techniques.

In the discussion, the authors bring up a short example of the characterisation of crystal defects in nanowires. This does somewhat break up the text, which is largely focused on the archaeological object. However, I accept that it does present another application of the technique, and one that may be relevant to a broader spectrum of potential end-users. As such, I agree to its inclusion in this proof of concept paper.

b) The earliest lost-wax cast metal object

Together with the presentation of USD-PL as a great development in materials characterisation, this manuscript constitutes the first detailed report on the object that constitutes the earliest evidence of lost-wax casting in the world. This small, wheel-shaped 'amulet' had been presented previously at conferences and mentioned in passing in a few publications, but it had not been published in any detail. This publication was therefore long awaited and it will find many interested readers among archaeologists and historians interested in ancient technologies.

The ghost structure revealed by USD-PL is testament to the previous existence of a eutectic phase of cuprite dendrites in a matrix of metallic copper spanning the whole object. The latter phase subsequently corroded, hence the bulk of the sample is made up of copper oxides (and other corrosion products), but the microstructure is diagnostic of its former presence. Such a phase could only have formed on molten copper, and the lack of any crystal deformation shows that the object was cast with very little if any subsequent work on the object, such as hammering or soldering. The authors' thorough and convincing explanation of the object manufacture and subsequent corrosion, accompanied by an excellent illustration in Fig 4, is based on sound knowledge of metal casting and weathering mechanisms and most likely represent a truthful summary of the artefact's life-history. All in all, the results thus confirm conclusively that the object was cast. It should be noted, however, that USD-PL alone does not demonstrate that the object would have been cast by the lost-wax technique. The inference that lost-wax was the technique employed is based primarily on the shape and macroscopic examination of the object, namely its three-dimensionality. As a matter of fact, the extended legend to Figure 1 presented as supplementary material argues quite convincingly that the object must have been manufactured by this method without any reference to the USD-PL imaging results ("no plane intercepts jointly the equatorial symmetry planes of the support ring and of the spokes without inducing an undercut").

This point should be made more explicit in the paper, as it might otherwise convey to untrained archaeologists the impression that USD-PL can help distinguish between objects cast by lost-wax and those cast using other techniques, such as piece moulds. In this case at least, the differentiation is based on other lines of evidence, including its coherence with later history of metallurgy in the same area (and indeed documented in other finds at the site) that show evidence of lost-wax casting. If this same object had been found in China, where there is a long tradition of casting extremely intricate shapes using piece-moulds rather than lost wax, we would probably be more cautious before concluding the use of lost-wax (e.g. Zhou and Huang 2015). Having said this, and all things considered, I fully concur with the authors that lost-wax casting is the more parsimonious explanation for this object.

Discussion

The main archaeological significance of this study is that it pushes back the origins of lost-wax casting and thus allows us to revisit the origins of this technique as a technological innovation, and to place it more broadly in the history of metalworking techniques. Many archaeologists and anthropologists are interested in the social and economic contexts that trigger the emergence of innovations, and this study provides an excellent case in point for discussion in this field. While I acknowledge the challenge of space constraints, I would encourage the authors to expand discussion of these issues (and the editors to allow for such expansion), at least to point to further implications of this find to the broader field. If such discussion is reduced to the minimum, then the paper loses relevance to archaeological readership beyond its anecdotal status as 'the earliest'.

First, it would be useful to know more about the specific archaeological context where this object was retrieved, as well as any additional information available about coeval metal artefacts found at the site. The authors note the singularity of the object being made of unalloyed copper, as opposed to the arsenical copper that is typical of the region and period. At this juncture, they refer to the work by Thornton et al. on metal analyses from a different site in the region. I assume they have to do this owing to the lack of analyses of other objects from Mehrgarh, but it would be useful to have this point confirmed in the paper.

Second, the brief discussion focuses on the diachronic evolution of lost-wax casting, and the authors argue that subsequent experimentation would have led metallurgists to realise the advantages of using alloys for lost-wax casting, as opposed to unalloyed copper. While I agree that this is a likely scenario, I think the discussion implicitly assumes a unilinear evolutionary trajectory for this technique that would have been driven by efficiency. In other words, that lost-wax casting emerged as an innovation to fulfill the technical need to cast intricate forms, and that later developments would have been further steps in the same direction. This assumption is prevalent in studies on the early history of lost-wax casting (e.g. Davey 2009) and many studies on early technologies generally.

In my view, it is significant that the wheel-shaped artefact in question does not fully exploit the potential of lost-wax casting to achieve fully three-dimensional shapes. Bar for some small superimpositions at the joints, the object is practically flat, and arguably a very similar one could have been cast more easily using an open mould. In a recent paper focused on the discovery of lost-wax casting in South America (Martín-Torres and Uribe-Villegas 2015), a colleague and I argued that the emergence of this technique could have been triggered by experimentation with beeswax as a ritually important material, and that the use of wax may have been required by ritual prescription rather than by technical need. This argument was supported by several strands of evidence, including the use of lost-wax casting for artefacts where there was no technical need for this method, and ethnographic evidence for the symbolic significance of wax. Given the Mehrgarh object is seemingly non-utilitarian, and that lost-wax casting does not appear to have been an utmost technical requirement to achieve the desired shape, I would be inclined to support a similar explanation for the adoption of this technology here. In any case, it would be useful to see the authors of this manuscript engaging with this discussion, given the high relevance of their find to the subject. More broadly, it would be interesting to bring up any additional evidence there may be for beekeeping, and the use of bees products (or alternative materials) in the period and region under consideration.

In sum, this is a rigorous and highly interesting paper on both methodological and archaeological grounds, and I recommend that it is published, subject to a slight 'toning down' of the role of USD-PL in 'proving' that the object was cast by the lost-wax technique, and some elaboration on the archaeological context of the object and its significance to broader discussion of the emergence of lost-wax casting, and of archaeological innovations in general.

Minor comments, in no particular order:

Title: I would change "exemplified on a 6000.." to "exemplified on the earliest metal object cast by lost wax". This will make the subject more obvious to potential readers.

Indicate the country where Mehrgarh is located

Figure 4 is excellent but I would suggest two modifications: first (and more importantly), the liquid being poured out of the crucible in b(1) should be coloured yellow, as per their legend (Cu is yellow, red would be Cu₂O); second (and less important), the crucible shape could be made shallower, in keeping with the typical shape of early crucibles.

Native copper: Ag, Au and Hg are mentioned as trace elements in the metal (in addition to Fe, which could easily derive from soil contamination). A few paragraphs later it is asserted that the object could be made of "possibly native copper". However the authors draw no explicit links between these two pieces of evidence. While acknowledging the non-quantitative nature of the data and the corroded condition of the sample, the absence of other elements such as Co, Mn or Pb, and the traces

mentioned, would seem indeed consistent with native copper (see e.g. Pernicka et al. 1997).

References

- Davey, C.J., 2009. The early history of lost-wax casting, in *Metallurgy and Civilisation: Eurasia and Beyond*, eds. J. Mei and Th. Rehren. London: Archetype, 147-54.
- Martinón-Torres, M., Uribe-Villegas, M.A., 2015. Technology and culture in the invention of lost-wax casting in South America: an archaeometric and ethnoarchaeological perspective. *Cambridge Archaeological Journal*, 25, 377-390.
- Pernicka, E., Begemann, F., Schmitt-Strecker, S., Todorova H. and Kuleff, I. 1997. Prehistoric copper in Bulgaria: its composition and provenance. *Eurasia Antiqua*, 3, 41-180.
- Zhou, W., and Huang, W. 2015. Lost-wax casting in ancient China: new discussion and old debates. *JOM*, 67, 1629-1636.

Reviewer #3 (Remarks to the Author):

While the authors present an admirable and scientifically rigorous forensic archeological investigation of a 6000 year old amulet using photoluminescence microscopy, their coining of the term 'ultimate spatial dynamics-photoluminescence imaging' to describe what appears to be a relatively straightforward application of PL imaging borders on hyperbole. They repeatedly quote 100 nanometer spatial resolution in the text of the article, but fail to actually demonstrate spatial resolution anywhere near this magnitude. While Fig.5 showing PL from ZnO nanowires may achieve resolution approaching 500 nm's, none of the images of the amulet demonstrate or require sub-micron resolution.

While the authors refer to gigapixel luminescence imaging based on coupling of full-field imaging and optimized raster scanning and refer to an ability to switch between conventional and synchrotron sources to provide unprecedented tunability of excitation wavelength, they fail to provide any specific description of their 'ultimate spatial dynamics PL imaging' methods. Based on the prominence of this term in the title, abstract, and introduction one is forced to conclude that this approach, described as a 'paradigm shift for characterization' and 'a breakthrough methodology to study nano- and polycrystalline materials,' is the major claim of this work.

Given that this revolutionary new method is the primary claim of this work, there must be much more detail provided about the method and demonstration of its performance. Measurement of the optical point spread function across a suitable nano-scale object would enable the authors to quantitatively demonstrate 100-nm scale spatial resolution. Description of the hardware and methodology would enable others to understand and hopefully reproduce and validate the approach of this potential paradigm shift in characterization methods.

Considering the rapid and substantial progress being made in the field of super-resolution imaging beyond the diffraction limit, such as the 2014 Nobel Prize in chemistry for super-resolved fluorescence microscopy, the claims of a paradigm shift and breakthrough methodology set a high bar that this work does not meet.

The scientific work to understand the metallurgical process to produce the amulet is admirable and beyond the expertise of this reviewer. This paper may well deserve publication in this journal based on the investigation of this amulet. On the aspect of 'ultimate spatial dynamics PL imaging' the paper requires major modification before it can be considered for publication.

REVIEWERS' COMMENTS:

Reviewer #1 (Remarks to the Author):

The authors have concisely and convincingly addressed all the comments raised by this reviewer. The paper is clearer and stronger, and its broader significance is well articulated. I can now recommend publication without reservations.

Incidentally, the authors' reply to my comments mentions in passing that they were reluctant to use SEM extensively in the sample because of concerns that it might affect its photoluminescence. I think this is an important point that should be mentioned in the paper, to avoid problems in future research. Otherwise scholars might spend too much time analysing a corroded sample an SEM, as per normal practice, before considering the use of photoluminescence.

Marcos Martinon-Torres

Reviewer #3 (Remarks to the Author):

I agree with the authors that the range of spatial resolution provided by high spatial dynamics imaging is key in detecting the patterns that allow them to explain the metallurgical origin of the amulet. I disagree that they have demonstrated diffraction-limited resolution on the order of 400 nm.

I commend the authors for adding Fig.9 in the supplemental materials, showing a PL image of a 400-nm diameter CdS nanocrystal. I am puzzled by their statement that this figure constitutes a measurement of the point spread function of their imaging system. There is no evaluation of the width of the transition across the edge of the nanocrystal. The authors have made no attempt at a quantitative evaluation of the PSF.

The authors correctly calculate the theoretical, diffraction-limited resolution for a NA=1.25 objective at 935 nm to be 374 nm. They go on to state that the projected pixel size of the detector is 154 nm. While these are necessary conditions to achieve such resolution, they are not sufficient conditions to achieve that resolution. Achieving diffraction-limited resolution requires correction for spherical aberration that occurs in the medium being imaged. Proving diffraction-limited resolution requires experimental proof, not simply calculating what the theoretical limit is.

The authors also added Fig.10 showing the microstructure of the eutectic pattern. From the scale bar in the figure it is clear that the features in this image are ~ 1 μm in diameter, yet the authors state that this image demonstrates ca. 400-nm resolution. As in Fig.9, this statement puzzles me.

In the text, line 126 states "350-nm lateral resolution allows clear observation of rod-like structure of high-yield luminescent Cu_2O ," but the scale bar of 100 μm indicates that the rod-like structures in the interdendritic space are 1-2 μm or greater in lateral size. These are the smallest features in this image; hence 350-nm resolution is neither demonstrated nor needed to make this critical identification of these features in the amulet.

Based on these observations I do not believe that sub-micron resolution is required to explain the origins of the amulet. In addition, the authors have failed to substantiate their claims of diffraction-limited resolution. The authors themselves state "attaining ultimate spatial resolution is neither the target nor a critical component of this work."

I suggest that the authors modify the manuscript to delete statements of diffraction limited resolution and delete statements of 374-nm or ca. 400-nm resolution. It would be appropriate to state ca. 1 μm resolution. If they are not comfortable with these changes then they must provide quantitative experimental evidence of the higher resolution.

Reviewers' comments:

First, we would like to thank the two referees of our work for extremely valuable comments.

We have made our best to thoroughly consolidate and improve our manuscript by taking into account these useful and inspiring suggestions and corrections, including by integrating novel data from complementary characterisation.

Reviewer #1 (Remarks to the Author):

This manuscript presents a well-written report on a rigorous and original analytical study of what arguably represents the earliest metal object cast using the lost-wax technique. The paper has two main aims:

a) to present ultimate spatial dynamics-photoluminescence imaging (USD-PL) as a novel technique with great potential for the characterisation of bulk heterogeneous materials at multiple scales;

b) to demonstrate that a small, corroded object recovered in an early context at the site of Mehrgarh was manufactured by the lost-wax casting method, and therefore represents the earliest evidence for the use of this technique.

Both of the above points are covered convincingly and are significant for a wide range of disciplines, from materials science through to archaeology and archaeometallurgy. Therefore I believe the paper merits publication in Nature Communications, although I recommend a few revisions. These are noted below, together with some more general comments.

a) USD-PL

Although I have experience in the analysis of archaeological metal objects (including some highly corroded like the one discussed in this paper), and have used the other techniques employed in this project (optical microscopy, Raman and SEM), I do not have direct experience with photoluminescence spectroscopy. Hence other reviewers will be more qualified to comment on the merits of this technique, and the suitability of the protocols followed.

The main breakthrough demonstrated in this study is the step forward from traditional PL analysis enabled by the generation of gigapixel luminescence images; this is obtained by combining full-field imaging with optimised raster scanning, and in turn it allows analysts to map samples at successive length scales across several orders of magnitude, from micrometers to centimetres. The authors describe this as a 'paradigm shift'. While only time and future tests will tell whether USD-PL truly brings a new paradigm, it is safe to say that this development does increase very noticeably the versatility and potential applications of the technique.

We agree with the referee that the development of USD-PL increase noticeably the versatility and potential applications of luminescence imaging, in particular to heterogeneous materials. We also agree on the fact that the term 'paradigm shift' may read performative rather than informative (this point was also raised by the second referee). We have therefore opted for deleting this term while providing a more detailed description of the methodological developments made and of the performances attained.

The main demonstration of the proof of concept is the study of the corroded copper object from Mehrgarh.

Figure 2 beautifully illustrates how USD-PL allows the analyst to see ghost microstructural features that would be invisible under conventional light microscopy. The ghost eutectic structure of the non-stoichiometric copper oxides is revealed distinctively, and proves beyond reasonable doubt that the object was cast in one piece. While the results are thus quite convincing, it would be interesting to compare these two sets of images (optical microscopy and USD-PL) to images of the same area obtained under an SEM. The latter is a technique commonly employed by archaeometallurgists to examine ghost microstructures in corroded objects, and one is left wondering if the features visible by PL would not have been observed under an SEM too. Admittedly, an SEM image is unlikely to be as impressive as that of PL, not least because of the contrast in luminescence that would not be perceptible by a BSE or SE detector on an SEM. However, a comparison between SEM and USD-PL would perhaps allow a more realistic graphic assessment of the differences between both techniques.

We agree with the referee on the benefit of comparing SEM images with USD-PL and optical microscopy images. We have therefore added a corresponding new figure (see **Supplementary Fig. 8**). These images further demonstrate that, due to the uniform composition in cuprous oxide across the corroded eutectic, no contrast and thus no eutectic pattern are observed in BSE/SE detection.

Please note that we were reluctant to image a very extended portion of the artefact using systematic scanning electron microscopy as radiation damage induced by the SEM electron beam might impact the photoluminescence response for further characterisation.

In the discussion, the authors bring up a short example of the characterisation of crystal defects in nanowires. This does somewhat break up the text, which is largely focused on the archaeological object. However, I accept that it does present another application of the technique, and one that may be relevant to a broader spectrum of potential end-users. As such, I agree to its inclusion in this proof of concept paper.

b) The earliest lost-wax cast metal object

Together with the presentation of USD-PL as a great development in materials characterisation, this manuscript constitutes the first detailed report on the object that constitutes the earliest evidence of lost-wax casting in the world. This small, wheel-shaped 'amulet' had been presented previously at conferences and mentioned in passing in a few publications, but it had not been published in any detail. This publication was therefore long awaited and it will find many interested readers among archaeologists and historians interested in ancient technologies.

The ghost structure revealed by USD-PL is testament to the previous existence of a eutectic phase of cuprite dendrites in a matrix of metallic copper spanning the whole object. The latter phase subsequently corroded, hence the bulk of the sample is made up of copper oxides (and other corrosion products), but the microstructure is diagnostic of its former presence. Such a phase could only have formed on molten copper, and the lack of any crystal deformation shows that the object was cast with very little if any subsequent work on the object, such as hammering or soldering.

The comment from the referee is indeed particularly appropriate and we have added a sentence indicating that such a phase could only have formed on molten copper, and that the lack of any crystal deformation shows that the object was made with very little if any subsequent work on the object.

The authors' thorough and convincing explanation of the object manufacture and subsequent corrosion, accompanied by an excellent illustration in Fig 4, is based on sound knowledge of metal casting and weathering mechanisms and most likely represent a truthful summary of the artefact's life-history.

All in all, the results thus confirm conclusively that the object was cast. It should be noted, however, that USD-PL

alone does not demonstrate that the object would have been cast by the lost-wax technique. The inference that lost-wax was the technique employed is based primarily on the shape and macroscopic examination of the object, namely its three-dimensionality. As a matter of fact, the extended legend to Figure 1 presented as supplementary material argues quite convincingly that the object must have been manufactured by this method without any reference to the USD-PL imaging results ("no plane intercepts jointly the equatorial symmetry planes of the support ring and of the spokes without inducing an undercut").

This point should be made more explicit in the paper, as it might otherwise convey to untrained archaeologists the impression that USD-PL can help distinguish between objects cast by lost-wax and those cast using other techniques, such as piece moulds.

We thank the referee for this important remark. We have done our best to clarify in several parts of the paper, in particular in the abstract and discussion, the fact that geometrical considerations show that the lost wax technique was used, while USD-PL allows elucidating the full operating sequence and indicates that the artefact was cast in a single piece.

In this case at least, the differentiation is based on other lines of evidence, including its coherence with later history of metallurgy in the same area (and indeed documented in other finds at the site) that show evidence of lost-wax casting. If this same object had been found in China, where there is a long tradition of casting extremely intricate shapes using piece-moulds rather than lost wax, we would probably be more cautious before concluding the use of lost-wax (e.g. Zhou and Huang 2015). Having said this, and all things considered, I fully concur with the authors that lost-wax casting is the more parsimonious explanation for this object.

We agree with the referee and have modified the manuscript accordingly, indicating that our findings are "in agreement with the history of metallurgy in Balochistan that shows evidence of an important development of lost-wax casting as demonstrated by extraordinary finds during the Late Chalcolithic (end of the 4th mill. BC), and the absence of any tradition of casting intricate shapes using piece-moulds as for instance reported in China (e.g. Zhou and Huang, 2015)."

Discussion

The main archaeological significance of this study is that it pushes back the origins of lost-wax casting and thus allows us to revisit the origins of this technique as a technological innovation, and to place it more broadly in the history of metalworking techniques. Many archaeologists and anthropologists are interested in the social and economic contexts that trigger the emergence of innovations, and this study provides an excellent case in point for discussion in this field. While I acknowledge the challenge of space constraints, I would encourage the authors to expand discussion of these issues (and the editors to allow for such expansion), at least to point to further implications of this find to the broader field. If such discussion is reduced to the minimum, then the paper loses relevance to archaeological readership beyond its anecdotal status as 'the earliest'.

A full paragraph has been added concerning the context that triggered the emergence of lost-wax casting at Mehrgarh (see also below for more detail).

First, it would be useful to know more about the specific archaeological context where this object was retrieved, as well as any additional information available about coeval metal artefacts found at the site. The authors note the singularity of the object being made of unalloyed copper, as opposed to the arsenical copper that is typical of

the region and period. At this juncture, they refer to the work by Thornton et al. on metal analyses from a different site in the region. I assume they have to do this owing to the lack of analyses of other objects from Mehrgarh, but it would be useful to have this point confirmed in the paper.

Full details about archaeological context of the amulet are given in **Supplementary Fig. 1**, as well as for the coeval artefacts made by lost-wax casting and discovered at Mehrgarh, including images of each artefact. Except from these pieces, metal from the Early Chalcolithic levels is known via circa 100 very small fragments so badly preserved that it is impossible to identify either the type of corresponding objects (except 3 pins) or the metal composition. We know from other sites in the Balochistan area that arsenical copper was used during the early Chalcolithic period (on-going work). As these data remain unpublished so far, we refer to the work by Thornton et al. work on the Tepe Yahya metal collection.

Second, the brief discussion focuses on the diachronic evolution of lost-wax casting, and the authors argue that subsequent experimentation would have led metallurgists to realise the advantages of using alloys for lost-wax casting, as opposed to unalloyed copper. While I agree that this is a likely scenario, I think the discussion implicitly assumes a unilinear evolutionary trajectory for this technique that would have been driven by efficiency. In other words, that lost-wax casting emerged as an innovation to fulfill the technical need to cast intricate forms, and that later developments would have been further steps in the same direction. This assumption is prevalent in studies on the early history of lost-wax casting (e.g. Davey 2009) and many studies on early technologies generally.

In my view, it is significant that the wheel-shaped artefact in question does not fully exploit the potential of lost-wax casting to achieve fully three-dimensional shapes. Bar for some small superimpositions at the joints, the object is practically flat, and arguably a very similar one could have been cast more easily using an open mould. In a recent paper focused on the discovery of lost-wax casting in South America (Martín-Torres and Uribe-Villegas 2015), a colleague and I argued that the emergence of this technique could have been triggered by experimentation with beeswax as a ritually important material, and that the use of wax may have been required by ritual prescription rather than by technical need. This argument was supported by several strands of evidence, including the use of lost-wax casting for artefacts where there was no technical need for this method, and ethnographic evidence for the symbolic significance of wax. Given the Mehrgarh object is seemingly non-utilitarian, and that lost-wax casting does not appear to have been an utmost technical requirement to achieve the desired shape, I would be inclined to support a similar explanation for the adoption of this technology here. In any case, it would be useful to see the authors of this manuscript engaging with this discussion, given the high relevance of their find to the subject. More broadly, it would be interesting to bring up any additional evidence there may be for beekeeping, and the use of bees products (or alternative materials) in the period and region under consideration.

Artefacts from later periods in Balochistan indicates a rather linear evolutionary trajectory concerning the moulding technology, and a conservative attitude concerning the alloy: CuPb will remain dedicated to lost-wax casting as soon as it has been concocted, hollow casting appear at the end of the late 4th or beginning of the 3rd mill. BC (Leopards weight), hollow statuary with use of a clay core during the 3rd mill BC, see Mille et al 2005 and Mille et al 2013.

Mille, B., Bourgarit, D. & Besenval, R. 2005. Metallurgical Study of the "Leopards Weight" from Shahi-Tump (Pakistan). In: Jarrige, C. & Lefèvre, V. (eds.) South Asian Archaeology 2001. Paris: Editions Recherches sur les Civilisations, 237-244.

Roux, V., Mille, B. & Pelegrin, J. 2013. Innovations Céramiques, Métallurgiques Et Lithiques Au Chalcolithique : Mutations Sociales, Mutations

Techniques. In: Jaubert, J., Fourment, N. & Depaepe, P. (Eds.) Transitions, Ruptures et continuité en Préhistoire, XXVIIe Congrès Préhistorique de France, Bordeaux - Les Eyzies, 31 Mai - 5 Juin 2010. Société Préhistorique Française, 61-73.

A paragraph has been added on the symbolic significance of lost-wax casting at Mehrgarh. It is here suggested a transfer from the manufacture of clay figurines and amulets, discovered by hundreds from the Neolithic and Chalcolithic levels. An assumption has been made concerning the modelling of these religious /magical artefacts: since shaping clay or wax relies to the same know how, it is suggested that the manufacture of the wax models had to be performed by the same individuals in charge of the making of the clay amulets. Lost-wax casting was therefore the only way to achieve a metal artefact identical (in a ritual sense) to those formerly made of clay.

Concerning beekeeping, no direct evidence helps to discuss on the importance of bees products. Only later iconographic representations of bees are known, and from other sites in Balochistan (namely the Makran area). For the period 3400–2800 BC, the best evidence is the Leopards weight (!), for the period 2800–2600 BC, a frieze of bees decorates a ceramic bowl. Despite the obvious relationship between the iconography of the Leopards weight and its lost-wax manufacture, the distance in time and space with the Mehrgarh amulet is excessive to mention this point in the present paper.

In sum, this is a rigorous and highly interesting paper on both methodological and archaeological grounds, and I recommend that it is published, subject to a slight 'toning down' of the role of USD-PL in 'proving' that the object was cast by the lost-wax technique, and some elaboration on the archaeological context of the object and its significance to broader discussion of the emergence of lost-wax casting, and of archaeological innovations in general.

We thank the referee for his/her comment, and we have made our best to answer the referee's comments by significantly improving our manuscript on the points mentioned. In particular, we have clarified the role of USD-PL to describe the metallurgical process used to produce the artefact, and further elaborated on the emergence of lost-wax casting and archaeological innovation.

Minor comments, in no particular order:

Title: I would change "exemplified on a 6000.." to "exemplified on the earliest metal object cast by lost wax".

This will make the subject more obvious to potential readers.

Indicate the country where Mehrgarh is located

The title was changed along the indicated lines. The country where Mehrgarh is located (Pakistan) was added.

Figure 4 is excellent but I would suggest two modifications: first (and more importantly), the liquid being poured out of the crucible in b(1) should be coloured yellow, as per their legend (Cu is yellow, red would be Cu₂O); second (and less important), the crucible shape could be made shallower, in keeping with the typical shape of early crucibles.

These changes were implemented in the figure.

Native copper: Ag, Au and Hg are mentioned as trace elements in the metal (in addition to Fe, which could easily derive from soil contamination). A few paragraphs later it is asserted that the object could be made of "possibly native copper". However the authors draw no explicit links between these two pieces of evidence. While acknowledging the non-quantitative nature of the data and the corroded condition of the sample, the absence of

other elements such as Co, Mn or Pb, and the traces mentioned, would seem indeed consistent with native copper (see e.g. Pernicka et al. 1997).

The text was modified to take into account this very helpful suggestion.

References

Davey, C.J., 2009. The early history of lost-wax casting, in *Metallurgy and Civilisation: Eurasia and Beyond*, eds. J. Mei and Th. Rehren. London: Archetype, 147-54.

Martinón-Torres, M., Uribe-Villegas, M.A., 2015. Technology and culture in the invention of lost-wax casting in South America: an archaeometric and ethnoarchaeological perspective. *Cambridge Archaeological Journal*, 25, 377-390.

Pernicka, E., Begemann, F., Schmitt-Strecker, S., Todorova H. and Kuleff, I. 1997. Prehistoric copper in Bulgaria: its composition and provenance. *Eurasia Antiqua*, 3, 41-180.

Zhou, W., and Huang, W. 2015. Lost-wax casting in ancient China: new discussion and old debates. *JOM*, 67, 1629-1636.

Reviewer #3 (Remarks to the Author):

While the authors present an admirable and scientifically rigorous forensic archeological investigation of a 6000 year old amulet using photoluminescence microscopy, their coining of the term 'ultimate spatial dynamics-photoluminescence imaging' to describe what appears to be a relatively straightforward application of PL imaging borders on hyperbole. They repeatedly quote 100 nanometer spatial resolution in the text of the article, but fail to actually demonstrate spatial resolution anywhere near this magnitude. While Fig.5 showing PL from ZnO nanowires may achieve resolution approaching 500 nm's, none of the images of the amulet demonstrate or require sub-micron resolution.

While the authors refer to gigapixel luminescence imaging based on coupling of full-field imaging and optimized raster scanning and refer to an ability to switch between conventional and synchrotron sources to provide unprecedented tunability of excitation wavelength, they fail to provide any specific description of their 'ultimate spatial dynamics PL imaging' methods.

We would like first to thank the reviewer for the statement that we “present an admirable and scientifically rigorous forensic archaeological investigation of a 6000 year old amulet using photoluminescence microscopy”.

The comment of the referee regarding PL imaging appeals to two main points of discussion.

A single image collected at high resolution would certainly not have allowed coming to the ubiquitous detection of the complex patterning arising from eutectic solidification. Ubiquitous detection of the eutectic and dendritic patterns was fundamental here for identifying here the full metallurgical structure. Indeed in samples from other periods and contexts, we have observed local eutectic formation not extending to the whole artefact but reflecting local composition as well as metallurgical conditions during metalworking. Here, the critical observation arises from being able to describe the artefact in identical analytical conditions from 400 nm – 10 μm (eutectic / grain size), 10–100 μm scale (dendrites), and 100 μm – 1 cm (distribution of interdendritic spaces).

Regarding the spatial resolution attained and used, the theoretical limit for near infrared detection in the far-field PL imaging geometry used is estimated at $935/2 * 1.25 = 374$ nm (see below for confirmation that the setup is diffraction-limited from the measurement of the point spread function). As indicated by the referee, this is indeed in this resolution range that the information is collected and used. The new **Supplementary Fig. 10** shows that micrometer sized eutectic patterns that can be discriminated thanks to the spatial resolution that can be attained at this wavelength.

We have therefore modified several sections of the text:

- We have measured the point spread function and added it as **Supplementary Fig. 9**;
- We have clarified that the lateral resolution is 374 nm which is at the diffraction limit in the NIR detection range with far-field detection;
- We have clearly defined high spatial dynamics as “a high ratio between field of view and lateral resolution” in order to strengthen its fundamental benefit to tackle the characterization of heterogeneous materials;
- We have added a new subsection ‘Photoluminescence imaging’ in the ‘Methods’ section dedicated to the description of the setup that allowed attaining ultimate “spatial dynamics”.

To our knowledge, it is the first time that multispectral imaging is performed over 5 orders of magnitude in length scale with a continuously tunable setup from the deep UV to near infrared in emission, also coupling a synchrotron and conventional source in excitation. We think that the full potential is all the more demonstrated in this work as

application is shown to two different types of systems: an extraordinary find from the Ancient Near East, and semiconductor nanowires studied in the context of novel processes of quality control. This brings new paradigms even for scientific fields which have never employed advanced photoluminescence imaging, such as Archaeology.

Based on the prominence of this term in the title, abstract, and introduction one is forced to conclude that this approach, described as a 'paradigm shift for characterization' and 'a breakthrough methodology to study nano- and polycrystalline materials,' is the major claim of this work.

Given that this revolutionary new method is the primary claim of this work, there must be much more detail provided about the method and demonstration of its performance. Measurement of the optical point spread function across a suitable nano-scale object would enable the authors to quantitatively demonstrate 100-nm scale spatial resolution. Description of the hardware and methodology would enable others to understand and hopefully reproduce and validate the approach of this potential paradigm shift in characterization methods.

Considering the rapid and substantial progress being made in the field of super-resolution imaging beyond the diffraction limit, such as the 2014 Nobel Prize in chemistry for super-resolved fluorescence microscopy, the claims of a paradigm shift and breakthrough methodology set a high bar that this work does not meet.

The scientific work to understand the metallurgical process to produce the amulet is admirable and beyond the expertise of this reviewer. This paper may well deserve publication in this journal based on the investigation of this amulet. On the aspect of 'ultimate spatial dynamics PL imaging the paper requires major modification before it can be considered for publication.

We think that our work provide significant conceptual innovation by putting the focus on “spatial dynamics” rather than spatial resolution. This concept has never been discussed and we think it is a further elaboration connected to the development of megapixel imaging that is applicable to many imaging and raster-scanning approaches.

However, we agree with the referee that the multiplication of the terms “paradigm shift” and “breakthrough methodology” is not serving the interest of the article and the concept of “spatial dynamics” that we expose. We have therefore rephrased several part of the text in order to better reflect the interest of the work and applications as underlined by both referees, as well as strengthening its application potential.

We have measured the experimental Point Spread Function by selecting CdS crystals and collecting data in identical conditions as the amulet. The corresponding figure is printed in **Supplementary Fig. 9**. This figure demonstrates that diffraction-limited imaging is obtained in the near infrared. For the near infrared range used, the theoretical spatial resolution limited by diffraction is $935/2 \times 1.25 = 374$ nm. Resolving structures at this limit requires a detector with a projected pixel of half this value, i.e. 187 nm. This value is attained here, as the projected pixel size of the detector is 154 nm with the optical elements implemented.

Regarding resolution, we would like to underline that attaining ultimate “spatial resolution” is neither the target nor a critical component of the present work, even though the setup is diffraction-limited. Indeed, it was not in our aim to claim the highest spatial resolutions as super-resolution approaches have triggered developments “at the expense of narrow fields of view and stringent requirements in sample surface roughness and slope.” Our work exploits critical improvements implemented at a synchrotron beamline to jointly optimise the full tunability of the setup and maintain state-of-the-art spatial resolution. We have therefore clarified in the manuscript that the ability to collect high signal-to-noise ratio photoluminescence data over 5 orders of magnitude in length scale holds the key of the significant results obtained, and present in more detail the hardware and methodology.

We agree with the referee that the interpretation of the amazing data retrieved from the amulet is a critical and

very unusual scientific result leading to deciphering the full metallurgy on a completely corroded artefact 6 millennia after burial which is pivotal in the history of Metallurgy.

Based also on the advice from Referee 1, we have therefore further expanded the depiction of the results that allowed deciphering the metallurgical process to produce the amulet. On the other hand, the information collected on the nanowires is equally surprising and demonstrates the capacity to tackle the heterogeneity of entire real systems that may contain otherwise hidden information at the crystal grain size.

Response to Reviewers' comments:

Reviewer #1 (Remarks to the Author):

The authors have concisely and convincingly addressed all the comments raised by this reviewer. The paper is clearer and stronger, and its broader significance is well articulated. I can now recommend publication without reservations.

Incidentally, the authors' reply to my comments mentions in passing that they were reluctant to use SEM extensively in the sample because of concerns that it might affect its photoluminescence. I think this is an important point that should be mentioned in the paper, to avoid problems in future research. Otherwise scholars might spend too much time analysing a corroded sample an SEM, as per normal practice, before considering the use of photoluminescence.

We thank the referee for his/her very useful comments.

Reviewer #3 (Remarks to the Author):

I agree with the authors that the range of spatial resolution provided by high spatial dynamics imaging is key in detecting the patterns that allow them to explain the metallurgical origin of the amulet. I disagree that they have demonstrated diffraction-limited resolution on the order of 400 nm.

We have deleted all mentions of a "diffraction-limited resolution on the order of 400 nm".

I commend the authors for adding Fig.9 in the supplemental materials, showing a PL image of a 400-nm diameter CdS nanocrystal. I am puzzled by their statement that this figure constitutes a measurement of the point spread function of their imaging system. There is no evaluation of the width of the transition across the edge of the nanocrystal. The authors have made no attempt at a quantitative evaluation of the PSF.

We are aware that the diameter of the particle compared to the projected pixel size is slightly too large to measure a rigorously usable experimental PSF to evaluate the highest spatial resolution attainable. This is particular due that we sought a non-conventional standard system (CdS crystals) that could be excited and which emission (in the near IR) could be detected in the exact working conditions used to study the amulet.

The authors correctly calculate the theoretical, diffraction-limited resolution for a NA=1.25 objective at 935 nm to be 374 nm. They go on to state that the projected pixel size of the detector is 154 nm. While these are necessary conditions to achieve such resolution, they are not sufficient conditions to achieve that resolution. Achieving diffraction-limited resolution requires correction for spherical aberration that occurs in the medium being imaged. Proving diffraction-limited resolution requires experimental proof, not simply calculating what the theoretical limit is.

The authors also added Fig.10 showing the microstructure of the eutectic pattern. From the scale bar in the figure it is clear that the features in this image are ~ 1 um in diameter, yet the authors state that this image demonstrates ca. 400-nm resolution. As in Fig.9, this statement puzzles me.

Please see our comment above.

In the text, line 126 states "350-nm lateral resolution allows clear observation of rod-like structure of high-yield luminescent Cu₂O," but the scale bar of 100 um indicates that the rod-like structures in the interdendritic space are 1-2 um or greater in lateral size. These are the smallest features in this image; hence 350-nm resolution is neither demonstrated nor needed to make this critical identification of these features in the amulet.

Based on these observations I do not believe that sub-micron resolution is required to explain the origins of the amulet. In addition, the authors have failed to substantiate their claims of diffraction-limited resolution. The authors themselves state "attaining ultimate spatial resolution is neither the target nor a critical component of this work."

I suggest that the authors modify the manuscript to delete statements of diffraction limited resolution and delete statements of 374-nm or ca. 400-nm resolution. It would be appropriate to state ca. 1 um resolution. If they are not comfortable with these changes then they must provide quantitative experimental evidence of the higher resolution.

We thank the referee for his/her comments and have indicated that the attained resolution is "ca. 1 um resolution". As recalled by the referee from a previous answer we made, attaining ultimate spatial resolution is neither the target nor a critical component of this work.

We would like to commend our two referees for the very useful comments that were made on our work.